# Identification of predicate creep under the 510(k) process: A case study of a robotic surgical device

**Charlotte Lefkovich[1]¤, Sandra Rothenberg[2]\***

1 Department of Public Policy, Rochester Institute of Technology, Rochester, New York, United States of America, 2 Department of Management and Public Policy, Rochester Institute of Technology, Rochester, New York, United States of America

¤ Current address: Southington, Connecticut, United States of America

\* slrbbu@rit.edu

**Data Availability Statement:** All 510K summary files used in this research are available from the 510K database at https://www.accessdata.fda.gov/scripts/cdrh/cfdocs/cfpmn/pmn.cfm

## Abstract

The FDA's 510(k) process for medical devices is based on "substantial equivalence" to devices clearedpre-1976 or legally marketed thereafter, known as predicate devices. In the last decade, several high-profile device recalls have drawn attention to this regulatory clearance process and researchers have raised questions about the validity of the 510(k) process as a broad clearance mechanism. One of the issues raised is the risk of predicate creep, a cycle of technology change through repeated clearance of devices based on predicates with slightly different technological characteristics, such as materials and power sources, or have indications for different anatomical sites. This paper proposes a new way to identify potential "predicate creep" through the use of product codes and regulatory classifications. We test this method by applying it to a case study of a Robotic Assisted Surgery (RAS) device, the Intuitive Surgical Da Vinci Si Surgical System. We find that there is evidence of predicate creep using our method, and discuss implications of this method for research and policy.

## Introduction

The United States Food and Drug Administration's [FDA's] 510(k) process for medical devices is based on "substantial equivalence" to devices cleared pre-1976 or legally marketed thereafter, known as predicate devices. In the last decade, several high-profile device recalls have drawn attention to this regulatory clearance process and researchers have raised concerns about the validity of the 510(k) process as a broad clearance mechanism. These concerns include, but are not limited to, a lack of a clear definition of "intended use," using recalled devices as predicates, lack of publicly available data regarding predicates and determinations of substantial equivalence, equating substantial equivalence to safety, and a lack of sufficient non-clinical data to support claims of substantial equivalence, among others. As outlined in the literature review below, there is a body of research that documents these problems and the resulting safety issues that have stemmed from them.

**Funding:** The author(s) received no specific funding for this work.

**Competing interests:** The authors have declared that no competing interests exist.

One of the specific concerns in the 510(k) process is the risk of predicate creep, a cycle of technology change through repeated clearance of devices based on predicates with slightly different technological characteristics, such as materials and power sources, or have indications for different anatomical sites [1, 2]. Predicate creep has been identified as one of the causes of some well-known medical device failure, such as the clearance of metal on metal hip implants, surgical meshes, and the ReGen Menaflex collagen scaffold, where what might seem as small differences between a device and its predicate were enlarged over time [1, 3, 4].

While 510 (k) process is intended to allow some level of innovation and thus technical change in new devices, the challenge is to identify when these changes are not "substantially equivalent" or are changes that may lead to safety issues. One of the most direct methods for identifying predicate creep is by making technical comparisons across a predicate ancestry tree. While perhaps the most accurate, this method is time consuming and is limited by the availability of data on the technical characteristics of the products. This paper proposes using product codes and regulatory classifications as an alternate means to identify predicate creep. The methods will be tested on a Da Vinci Si Surgical System, a robotic surgical platform. In this paper we will first review the existing research on concerns with the 510(k) process and research on predicate creep. Then, we review our methods to gather and analyze predicate data. We illustrate the use of this method on the Da Vinci Si Surgical System. We conclude with a discussion of the implications for our findings for policy and future research.

## Literature review

Many of the critiques of the 510(k) process relate to the notion of substantial equivalence. First, due to the lack of a clear official definition for the key terms "indications for use" and "intended use", the FDA has allowed permissive interpretation of these terms by applicants and inconsistent use of the terms across reviewers [5, 6]. Over time, this has resulted in the clearance of significantly altered devices, or even novel devices, as substantially equivalent to established predicates [5–7].

Second, the 510(k) process makes the implicit assumption that substantial equivalence means that a device is safe and effective, and that the predicates on which substantial equivalence determinations are based are safe. Substantial equivalence, however, only supports an assessment that the device introduces no new safety hazards and functions at least as effectively as the predicate device [8]. Moreover, if the predicate device poses risk or is ineffective, then the new device may perpetuate these flaws. It is not uncommon for a device to cite a recalled predicate [9–11], with one estimate being about 4.3% [11]. There is disagreement, however, as to whether this aspect of the process is invalid entirely or only for specific types of high risk or technologically complex devices [3, 9, 12].

Third, there are concerns regarding the use of multiple predicate devices for a single substantial equivalence determination. While most academics agree that single predicate submissions, which directly compare a new device to a single device on the market, provide significant assurance of safety and efficacy in most cases, many have raised questions about whether multiple and split predicates can provide the same level of assurance [2–4, 9].

Many of these concerns stem from an overarching issue that substantial equivalence allows a device to "piggyback" on the reasonable assurance of safety from existing predicate devices without undergoing independent testing [5, 12, 13]. These predicates can have also been cleared via substantial equivalence, creating a gap between the current device and the most recent device for which actual scientific evidence of safety was provided [14]. It is possible that, over multiple cycles of small device modifications and subsequent substantial equivalence findings, a new device may be cleared which is significantly dissimilar to the original predicate

for which scientific evidence exists [7]. This process is known as predicate creep [7, 9] and may lead to a device being introduced to the market which bears no resemblance to the original device.

Within the 510(k) process, requirements for non-clinical testing are used to mitigate the risk associated with small scale predicate creep, which in most cases works effectively. However, past research on predicate networks has highlighted the lack of publicly available non-clinical scientific data in substantial equivalence determinations. Both Zuckerman et al. [9] and Liebeskind et al. [15] used the FDA 510(k) Database and/or FOIA requests to trace the predicate history of different medical devices and found that there was a dearth of publicly available scientific data to support the claim of substantial equivalence. Liebeskind et al. [15], in particular, traced the predicate ancestry of robotic surgical systems, and found that only 19 (7.3%) 510(k) clearances provided clinical data, while 73 (27.9%) did not submit any supporting data. These studies suggest that non-clinical evidence to support substantial equivalence claims may be insufficient to ensure safety when there is predicate creep.

Past research on predicate creep primarily involves creating predicate ancestry trees and using direct technical comparisons of devices and their predicates. Such studies have identified issues concerning predicate creep in surgical meshes [4] and Pathwork Tissue of Origin Test [7], DePuy ASR XL Acetabular Cup System [3], power morcellators [16], among others. Ardaugh et al. [3], for example, studied the metal-on-metal hip implant over five decades with the purpose of identifying the cause of safety flaws present in the design. They identify that a unique combination of three characteristics in the ASR XL were approved through substantial equivalence of "split predicates," where you compare characteristics to different predicate devices. Since all three devices were deemed safe based on clinical trials and safe use of predicates in the consumer market, and the XL simply combined parts of the devices, it was placed on the market without undergoing clinical testing and the resulting product had to be recalled for particle shedding [3].

Direct technical comparisons can be challenging due to a lack of information, as discussed above, or the cumbersome size of predicate ancestry trees, particularly for newer devices. Using FOIA requests to attain more detailed information can also be onerous. This paper proposes another way to identify potentially important instances of predicate creep in a large predicate network through analysis of product classes and regulatory categorizations. We test this method on a well-known medical device, the Intuitive Surgical Da Vinci Si robotic surgical system.

## Method

The Da Vinci Si Surgical System is a robotic-based laparoscopic surgical tool initially approved by the FDA in 2000 which replaces a surgeon's hands with robotic arms for more precise control and motion [17]. While Intuitive has subsequently brought multiple iterations of the Da Vinci to market, it remains the only full RAS platform on the market as competitors struggle to develop a viable competitor around Intuitive's strong patent foothold [18]. The Da Vinci itself was initially approved under a 510(k) application based on a complex web of component-level substantial equivalence, most likely supplemented by additional testing. The Da Vinci Si was chosen because its function is well documented. Moreover, there is a non-negligible number of malfunctions that have occurred during the use of these types of devices [19].

Information on the devices within the Da Vinci Si Model predicate network collected through the 510(k) clearance database. We also searched other databases using the search function on the FDA website, which returns results from all FDA publications, including database information, conference presentations, regulations, and internal memos. Although this

method returns significantly more results, the search function offers limited filtering options and requires manual sorting to determine whether results are relevant. Information required for determination of substantial equivalence, such as predicate devices, intended use, indications for use, and scientific evidence may be presented in the summary page. The device summary page also includes the product classification code, which identifies a more device-specific classification based on the technological characteristics and intended use of a device. This can be used to identify potential predicate devices based on the substantial equivalence parameters for a new device. From this information, we created a database of devices that included devices that were predicates of or predicated on the device; we included information on the K#, device name, clearance date, product code, and any predicate or intended use information available for all devices relevant to the database construction parameter.

## Constructing predicate networks

The numbers of predicates in the Da Vinci Si predicate ancestry network made traditional tree diagrams too unwieldy, so an alternate diagram structure known as a network map was used to display predicate relationships within the clearance ancestry. After mapping the overall predicate network, we identify patterns in the use of existing mechanisms or structures which identify characteristics of the technology from a regulatory perspective. FDA product codes are particularly useful for this purpose, as they are designed to identify groups of devices with the same intended use and technological characteristics. Since possessing the same intended use is a requirement for clearance via substantial equivalence, it is likely that any device approved via 510(k) with a different product code that the predicate is of particular interest. The introduction of new product codes in the predicate ancestry tree should be indicative of the introduction of new technological characteristics or uses. While the device identification key is a set of functions or characteristics which distinguish devices in that product classification, we also looked at the regulatory description, which reflects the primary function or intended use as identified by the FDA.

## Data analysis

The network map for the Da Vinci Si is shown in Fig 1. Within this network mapping diagram each device is represented by a dot, with the dot size increasing based on the total number of substantial equivalence relationships that device is based on. Substantial equivalence relationships (also known as predicate-subject device relationships) are represented by lines drawn between the two devices involved. Each dot is labeled with the K# of the device which it corresponds to. The trace begins with the main subject device (i.e. the Da Vinci Si) on the left side of the trace, and advances toward the left, with each line originating at a subject device and terminating at its respective predicate. Therefore, the oldest devices present in the trace are located on the right side, although vertical alignment does not correlate exactly to clearance date, and the newest devices are to the left of the trace.

This network map included 2618 device instances (i.e. a device is cited as a predicate), with a total of 50 unique devices. The unique devices within this trace are classified under a total of 15 different product codes, with the majority of devices, including the various Da Vinci models, categorized under code NAY. Additional information about each device, including the manufacturer, clearance date, and any recalls issued, can be found in Table 1.

A list of the codes, the device identification key, and the regulatory description for each code is shown in Table 2 for the Da Vinci Si. The device identification key is a set of functions or characteristics which distinguish devices in that product classification, while the regulatory description is the primary function or intended use as identified by the FDA.

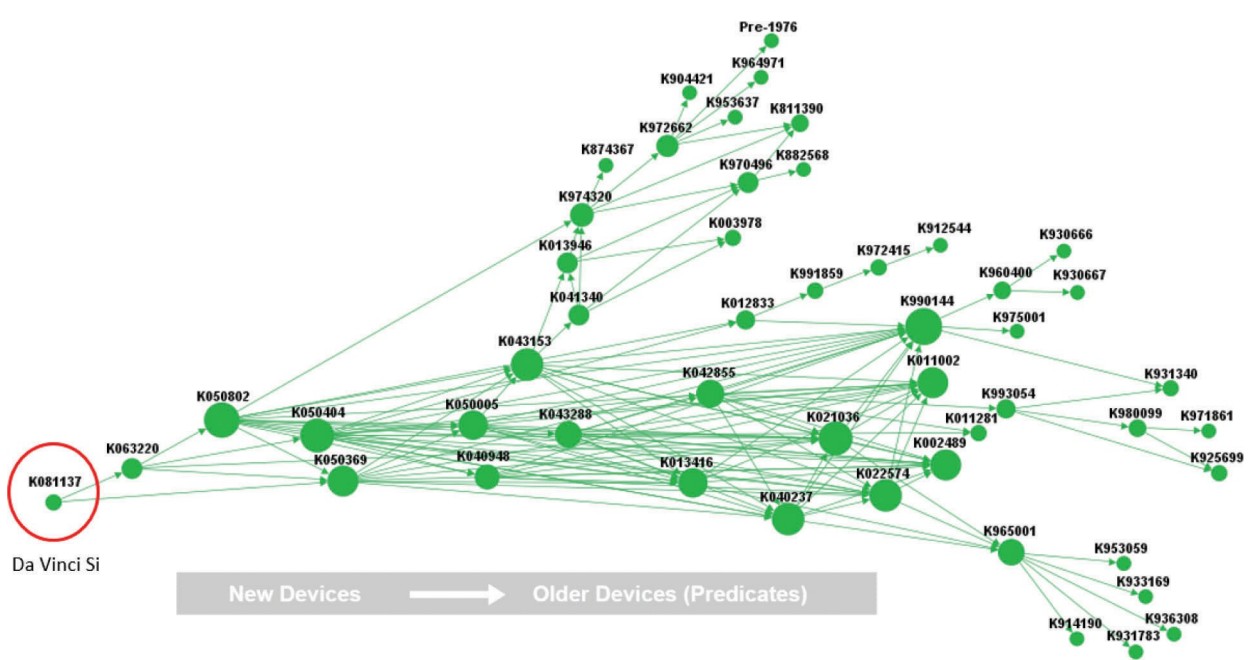

**Fig 1. Expanded Da Vinci Si predicate trace.**

A breakdown of the devices present in the trace color-coded by product code is pictured in Fig 2 below. The most prevalent code is NAY for computer controlled surgical systems. The next two most prevalent codes are GEH for cryosurgical units and GCJ for general laparoscopic surgery. Color-coding devices by code highlights common device functions within the trace and allows for easy identification of technical characteristics as they evolve through predicate generations.

Looking at the tree breakdown by product code in Fig 2, we can see the progressive evolution of devices from more general laparoscopic surgical tools (to the right) to the more technologically complex computer-controlled system of the Da Vinci Si. The five distinct branches emerging from the intertwined trace center are each dominated by one or two distinct product codes, while the devices within the central web belong almost exclusively to the same product classification as the Da Vinci Si. This implies that the technological characteristics the FDA uses to identify devices belonging to code NAY are a combination of the characteristics present in each distinct predicate group. This illustrates how larger "jumps" in technological complexity of new devices can occur through the 510(k) process, by combining the characteristics of multiple well-understood devices into a new type of device.

Another perspective to examine the technological evolution within the ancestral trace is based on the regulatory description rather than the product code. While the product code considers both the intended use and specific characteristics of a device, the regulatory description is a broader, more general description based on the function of the device as defined by the FDA. For example, the specific device description for product code GCJ is "laparoscope, general and plastic surgery," which specifies both a particular type of device and use, while the regulatory description "endoscope and accessories" specifies only a general classification of devices. Because of these broader descriptions, there is often overlap between the regulatory descriptions of different product codes. In this trace, devices from 15 product codes can be placed into 11 groups based on regulatory descriptions, resulting in the formation of two larger groups which contain the majority of devices within the trace.

**Table 1.** **Da Vinci Si predicates.**

| K Number | Device Name | Manufacturer | Clearance Date | Product Code | Recalls |
|---|---|---|---|---|---|
| K081137 | Intuitive Surgical Da Vinci Si Surgical System: Model Is3000 | INTUITIVE SURGICAL, INC. | 2/18/2009 | NAY | 24 –Class II |
| K063220 | Da Vinci S Surgical System-V1.1, Model Is2000 | INTUITIVE SURGICAL, INC | 12/1/2006 | NAY | 4 –Class II |
| K050802 | Modification To Intuitive Surgical Da Vinci Surgical System And Endoscopic Instruments | INTUITIVE SURGICAL, INC. | 6/29/2005 | NAY | 0 |
| K050369 | Intuitive Surgical Da Vinci Surgical System, Model Is2000 | INTUITIVE SURGICAL, INC. | 4/29/2005 | NAY | 43 –Class II |
| K050404 | Intuitive Surgical Da Vinci Surgical System And Endoscopic Instruments | INTUITIVE SURGICAL, INC. | 4/21/2005 | HET | 0 |
| K043288 | Modification To Intuitive Surgical Da Vinci Surgical System And Endoscopic Instruments | INTUITIVE SURGICAL, INC | 3/3/2005 | NAY | 0 |
| K050005 | Intuitive Surgical Monopolar Curved Scissors, Model 400179; Tip Cover Accessory, Model 400180 | INTUITIVE SURGICAL, INC. | 1/25/2005 | NAY | 2 –Class II |
| K043153 | Intuitive Surgical Da Vinci Surgical System And Endoscopic Instruments, Models Is1200 & Is1000 | INTUITIVE SURGICAL, INC. | 12/15/2004 | NAY | 0 |
| K042855 | Intuitive Surgical Harmonic Curved Shears Instrument | INTUITIVE SURGICAL, INC. | 11/12/2004 | NAY | 0 |
| K041340 | Guidant Microwave Ablation System | GUIDANT CORPORATION, CARDIAC SURGERY | 7/28/2004 | NEY OCL | 0 |
| K040237 | Intuitive Surgical Da Vinci Endoscopic Instrument Control System And Endoscopic Instruments | INTUITIVE SURGICAL, INC. | 7/7/2004 | NAY | 0 |
| K040948 | Intuitive Surgical Endopass Endoscopic Delivery Instrument, Model P/N 400170 | INTUITIVE SURGICAL, INC. | 5/5/2004 | NAY | 0 |
| K022574 | Intuitive Surgical Endoscopic Instrument Control System & Endoscopic Instruments, Model Da Vinci ISI 1000/1200 | INTUITIVE SURGICAL, INC. | 11/12/2002 | NAY | 0 |
| K021036 | Intuitive Surgical Da Vinci Surgical System, Model IS1200 | INTUITIVE SURGICAL, INC. | 6/26/2002 | NAY | 18 –Class II |
| K013946 | Flex 10 Accessory For The Afx Microwave Ablation System | AFX, INC. | 2/27/2002 | NEY OCL | 0 |
| K013416 | Intuitive Surgical Endowrist Endoscopic Instrument Family | INTUITIVE SURGICAL, INC. | 1/10/2002 | GEI | 0 |
| K012833 | Intuitive Surgical Bipolar Forceps | INTUITIVE SURGICAL, INC | 11/16/2001 | NAY | 4 –Class II |
| K011281 | Intuitive Surgical Ultrasonic Shears | INTUITIVE SURGICAL, INC. | 7/24/2001 | NAY LFL | 0 |
| K011002 | Intuitive Surgical Da Vinci Surgical System, Model Isi 1000 | INTUITIVE SURGICAL, INC. | 5/30/2001 | NAY | 0 |
| K003978 | Afx Microwave Generator, Flex Ablation Wand, Lynx Ablation Wand, Model Series 1000, P/N 102006, P/N 102007 | AFX, INC. | 5/22/2001 | NEY OCL | 0 |
| K002489 | Intuitive Surgical Da Vinci Endoscopic Control System | INTUITIVE SURGICAL, INC. | 3/2/2001 | NAY | 0 |
| K990144 | Intuitive Surgical Endoscopic Instruments, Intuitive Surgical Endoscopic Instrument Control System | INTUITIVE SURGICAL, INC. | 7/11/2000 | NAY | 3 –Class II |
| K993054 | Ultracision Harmonic Scalpel Coagulating Shears, Models Lcs-C5, Lcs-C1, Cs-23c, Cs-231, Cs-14c, Cs-141 | ETHICON ENDO-SURGERY, INC. | 12/9/1999 | LFL | 0 |
| K991859 | Dexide Bipolar Forceps Ii ** Device | UNITED STATES SURGICAL, A DIVISION OF TYCO HEALTHC | 6/23/1999 | GEI | 0 |
| K980099 | Ultracision Laparosonic Coagulating Shears (Lcs-5(Lcsk5 And Lcsb5)) | ETHICON ENDO-SURGERY, INC. | 4/9/1998 | LFL | 0 |
| K974320 | Cryogen Cardiac Cryosurgical System | CRYOGEN, INC. | 2/3/1998 | OCL | 0 |
| K972662 | Cryogen Cryosurgical System | CRYOGEN, INC. | 10/1/1997 | GEH | 0 |
| K972415 | Minisite*Bipolar Forceps** Device | UNITED STATES SURGICAL, A DIVISION OF TYCO HEALTHC | 9/19/1997 | GEI | 0 |
| K965001 | Intuitive Surgical Monarch Laparoscopic Manipulator | INTUITIVE SURGICAL, INC. | 7/31/1997 | GCJ | 0 |

*(Continued)*

**Table 1.** (Continued)

| K Number | Device Name | Manufacturer | Clearance Date | Product Code | Recalls |
|---|---|---|---|---|---|
| K971861 | Ultrasonic Hand Instruments | UNITED STATES SURGICAL, A DIVISION OF TYCO HEALTHC | 7/1/1997 | LFL | 0 |
| K970496 | Heartport Maze System: Cryoprobe Set | HEARTPORT, INC. | 5/9/1997 | OCL | 0 |
| K964971 | Cryogen Cryosurgical System | CRYOGEN, INC. | 3/28/1997 | GEH | 0 |
| K960400 | Diamond-Touch And Micro Diamond-Touch Instruments/ Diamond-Line Instruments/Diamond-Port (Access Parts) | SNOWDEN-PENCER | 3/12/1996 | FBM | 0 |
| | | | | GCJ | |
| | | | | GEI | |
| K953637 | CMS Accuprobe 550/530 | CRYOMEDICAL SCIENCES, INC. | 12/4/1995 | GEH | 0 |
| K953059 | Kittner Dissector | MEDICAL PERSPECTIVES CORP. | 9/14/1995 | GDY | 0 |
| K930666 | Reusable Laparoscopic Instruments W/ Electrocautery | SNOWDEN-PENCER | 5/19/1994 | GEI | 0 |
| K930667 | Reusable Laparoscopic Instruments | SNOWDEN-PENCER | 5/16/1994 | GCJ | 0 |
| K933169 | Inman Endoscopic Blunt Dissector | INMAN MEDICAL CORP. | 4/19/1994 | GCJ | 0 |
| K936308 | Endex Endoscopic Positioning System | ANDRONIC DEVICES, LTD. | 3/31/1994 | FQO | 0 |
| K931783 | AESOP (Automated Endoscopic System For Optimal Positioning) | COMPUTER MOTION, INC | 11/22/1993 | GCJ | 0 |
| K931340 | Grasp Forceps/Scissors/Needle Holder/Dissector | BAXTER HEALTHCARE CORP. | 7/1/1993 | GCS | 0 |
| K925699 | Harmonic Scalpel Laparosonic Clamp Coagulator Acc. | ULTRACISION, INC. | 5/17/1993 | GCJ | 0 |
| K914190 | Auto Suture(R) Endoscopic Fan Retractor | UNITED STATES SURGICAL, A DIVISION OF TYCO HEALTHC | 5/6/1992 | GAD | 0 |
| K912544 | Bipolar Forceps | EVEREST MEDICAL CORP. | 6/24/1991 | MAV | 0 |
| K904421 | CMS Oncoprobe | CRYOMEDICAL SCIENCES, INC. | 4/8/1991 | GEH | 0 |
| K882568 | 130 Cryo Unit & Assoc. Cryoprobes & Spray | SPEMBLY MEDICAL LTD. | 9/27/1988 | GEH | 0 |
| K874367 | Various Cardiac Cryoprobes Having Dia. & Cos. Diff | SPEMBLY MEDICAL LTD. | 1/4/1988 | HQO | 0 |
| K811390 | Ccs100 Cryosurgical System | FRIGITRONICS OF CONNECTICUT, INC. | 6/16/1981 | GEH | 0 |

**Table 2. Product codes in Da Vinci Si trace.**

| Code | # Devices | Device Identification | Regulatory Description |
|---|---|---|---|
| FBM | 1 | Cannula and Trocar, Suprapubic, Non-Disposable | Suprapubic urological catheter and accessories |
| NEY | 3 | System, Ablation, Microwave and Accessories | Electrosurgical cutting and coagulation device and accessories |
| NAY | 17 | System, Surgical, Computer Controlled Instrument | Endoscope and accessories |
| HET | 1 | Laparoscope, Gynecologic (And Accessories) | Gynecologic laparoscope and accessories |
| GEI | 5 | Electrosurgical, Cutting & Coagulation & Accessories | Electrosurgical cutting and coagulation device and accessories |
| LFL | 4 | Instrument, Ultrasonic Surgical | N/A |
| OCL | 5 | Surgical Device, For Cutting, Coagulation, And/Or Ablation of Tissue, Including Cardiac Tissue | Electrosurgical cutting and coagulation device and accessories |
| GEH | 6 | Unit, Cryosurgical, Accessories | Cryosurgical unit and accessories |
| GCJ | 6 | Laparoscope, General & Plastic Surgery | Endoscope and accessories |
| GDY | 1 | Gauze/Sponge, Internal, X-Ray Detectable | Nonabsorbable gauze for internal use |
| FQO | 1 | Table, Operating-Room, Ac-Powered | Operating tables and accessories and operating chairs and accessories |
| GCS | 1 | Endoscope, Battery-Powered and Accessories | Endoscope and accessories |
| GAD | 1 | Retractor | Manual surgical instrument for general use |
| MAV | 1 | Syringe, Balloon Inflation | Angiographic injector and syringe |
| HQO | 1 | Unit, Cautery, Thermal, Ac-Powered | Thermal cautery unit |

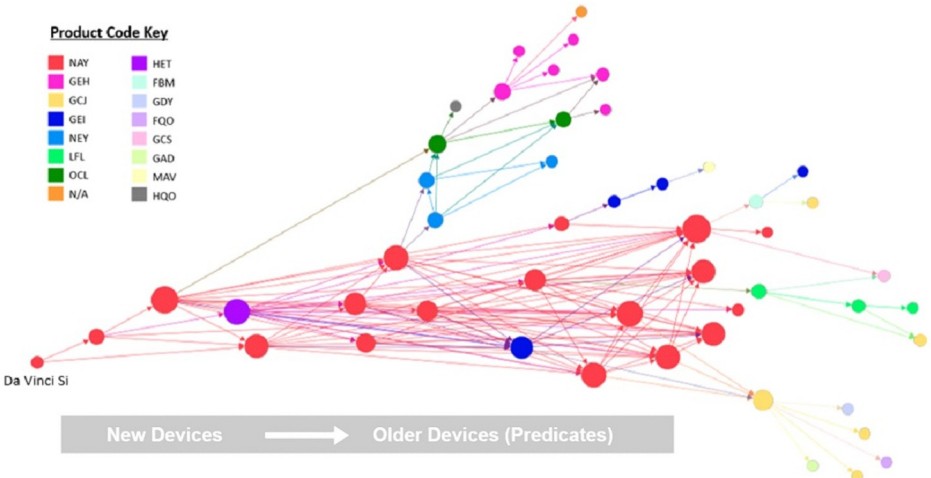

**Fig 2. Da Vinci Si trace color-coded by product classification code.**

The regulatory definition breakdown reveals that approximately half of the devices (24) within the trace are classified from a regulatory perspective as endoscopes and accessories. A further ~25% (13 total) are classified as electrosurgical cutting and coagulation devices, devices which use a high frequency electrical current to perform surgical operations. 6 of the devices are cryosurgical units, all classified under product code GEH, with the remaining devices representing a wide variety of functions and characteristics.

Unlike a grouping by product code, which highlights the evolution of specific technical characteristics over time as new codes are introduced to the trace, viewing the trace based on the regulatory description highlights groups of predicates based on general device functions. For example, in the product code trace the upper branch consists of four distinct product codes. However, when color-coded by regulatory description it becomes apparent that the originating predicate (branch tips) are all cryosurgical units with accessories, which serve as predicates to the electrosurgical cutting and coagulation devices that make up the middle of

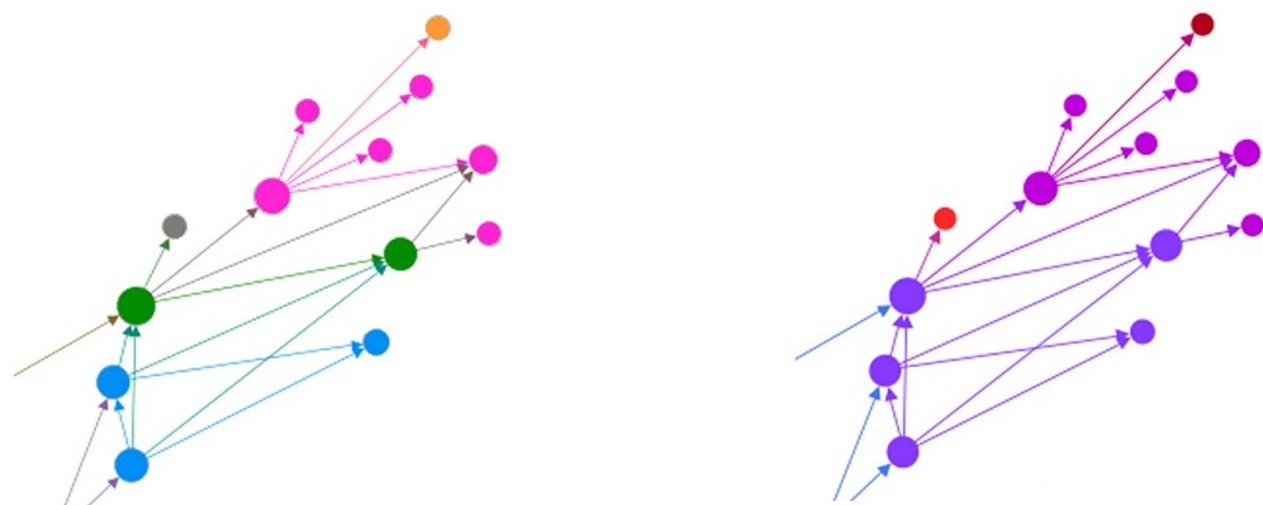

**Fig 3. Direct comparison of devices in upper Da Vinci Si predicate branch.**

the branch, which in turn serve as predicates for the computer controlled surgical devices present in the web center. Fig 3 shows a direct comparison of devices in the upper Da Vinci Si predicate branch, as shown in Fig 2. We can see from this comparison that while color-coding by the product code specifically identifies groups of devices which the FDA considers similar enough to be substantially equivalent, devices with the same intended use and technological characteristics, color-coding by regulatory description makes it easier to trace the general progression of technology over time based on the general function of these devices. The new central group present in the regulatory description trace combines two product codes to create a more general device grouping.

Fig 4 shows the devices in the entire DaVinci Si Trace color-coded by regulatory description. The teal nodes and lines represent devices approved as "endoscopic instruments and control systems" under code NAY. This style of coding based on regulatory description also draws attention to predicate devices with unusual functional characteristics that do not fit with the primary function of most devices within the trace. In some cases, this may be an indication of an unnecessary or ineffective predicate relationship, while in others it may be indicative of secondary device functions. For example, in this trace there is a device identified as a non-absorbable gauze/sponge for internal use. This device, the Medical Perspectives Kittner Dissector, is a sponge used during surgical procedures to prevent bleeding. It was identified as a predicate of the Monarch Laparoscopic Manipulator system, which included customized versions of many basic surgical instruments such as gauze. Although there is a purpose for including this device as a predicate for a custom surgical tool, this part of the system is secondary to the main function of the Da Vinci Si as a computer controlled surgical system.

Sorting predicates by regulatory description appears to allow for easy identification of both primary and secondary device functions, based on the prevalence and location of a given function in the trace. Primary device functions would be those identified directly by the regulatory description of the given device, while secondary functions would be functions present in predicate devices but absent in the regulatory description of the subject device. The more prevalent a function is in the predicate history, the more likely it is to be present in the subject device in at least a secondary capacity. Additionally, the significance of a particular secondary function to the overall function of the device appears to correspond to the number of predicates with that secondary function identified as a primary function. For example, the second most prevalent regulatory description in the Da Vinci Si trace is "electrosurgical cutting and coagulation device and accessories," which corresponds to the function of an essential component of the Da Vinci Si system. Although this is no longer listed as a primary function for the Da Vinci Si, it is an important component of the system.

## Discussion and conclusion

Given the purpose of the 510(k) process is to encourage incremental innovation in the medical device market by reducing regulatory barriers, while ensuring safety, it should be expected that there is a level of technology creep inherent in the process. If manufacturers are limited to only submitting identical devices for clearance through the 510(k) process, there can be no innovation. Even in new versions of existing devices submitted by the same manufacturer, such as the Da Vinci Models IS1000 and IS1200, there can be somewhat significant technological changes. Removing all technology creep from the process would only hinder progress within the medical devices industry. Thus, predicate creep in cases where the new device can be reasonably assured safe based on available scientific evidence, is beneficial to companies, patients, and regulators.

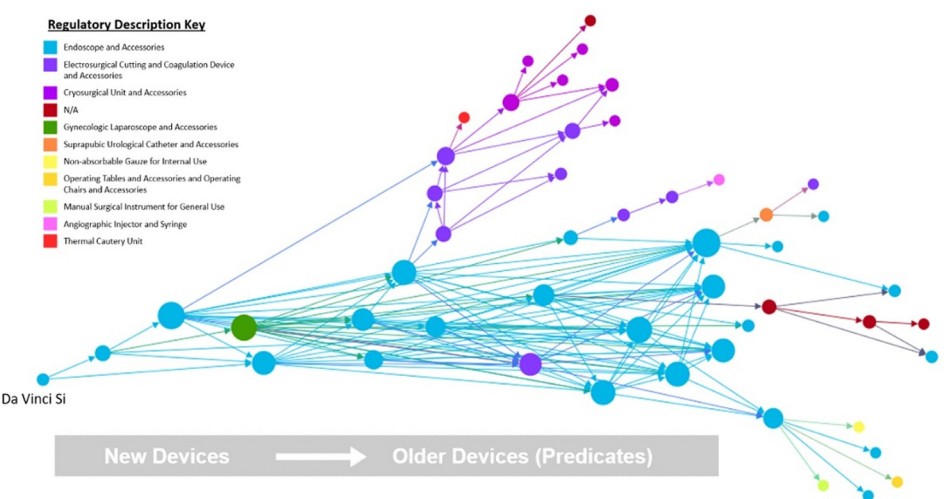

**Fig 4. Da Vinci Si trace with devices color-coded by regulatory description.**

This research focused specifically on examining the 510(k) clearance process as it was applied to Robotic Assisted Surgical systems, an emerging technology with a high degree of technical complexity, with specific focus on the Intuitive Da Vinci Si Surgical System. The objective of this research was to examine the predicate history of these devices and assess whether significant predicate creep by tracking changes and patterns in product codes and regulatory classifications.

The major contribution of this research to the discussion surrounding the 510(k) clearance process is the use of product codes and regulatory categorizations in predicate ancestry networks as a means to identify potentially problematic predicate creep. This method allowed us to see "jumps" in technological complexity of devices new devices made by combining the characteristics of multiple well-understood devices into a new type of device, such as with the NAY product code. We were also able to see the absorption of a primary predicate device function into a secondary system function, which is an indication of increasing complexity of devices over time, where the Da Vinci Si represents a particularly large leap in complexity. While the five main predicate branch groupings each generally contain one or two primary device functions that evolve and become more complex over time, the Da Vinci Si suddenly combines all those functions together into a single device where none of the functions serve as the primary function. The listed primary function "endoscope and accessories," is a very generalized term which only identifies the device as one used for internal imaging, even though it includes all the other secondary functions derived from the predicate devices, and is in fact used to directly perform surgeries.

## Limitations

This study was limited by several factors. First, the investigations performed in this study were based solely on data available publicly through FDA databases. Predicate devices are identified within the database only when application summaries are provided, which was not required for inclusion until the mid-2000's. This significantly limits the number of devices with traceable predicate histories. Moreover, the level of information contained within these summaries varies widely, and in many cases these summaries do not exist at all, due to the gradual evolution of requirements for 510(k) application submissions. We did consider requesting

additional data through FOIA, but decided against this route given that it would add a significant amount of time to the data gathering, with an FDA estimate of 18 to 24 months [20], and there was a precedent in prior research to rely primarily on information provided in the FDA website [9]. Additionally, given we are looking backwards in time, the fact that some predicates are missing actually makes any finding of predicate creep more significant.

Another limitation is that the information available through the FDA databases is usually from general device summaries, which are written to protect intellectual property rights, including minimal details about the specific technological characteristics of each device and the evidence provided to support equivalence claims. Thus, without the information to correlate findings to actual technological characteristics and the evidence used to support their existence, it is difficult to evaluate the significance of findings. Further, there is a lack of information regarding decision making of FDA officials during the review process creates a knowledge gap, which may explain why even large-scale technological creep was allowed through this accelerated regulatory processes. Lastly, we did not determine if the product classification code was made by the manufacturer or the FDA, or if the code change was driven by an administrative need or a technical feature.

Another limiting factor which may impact the significance of conclusions is the choice to limit the scope of this research to robotic surgical devices. Conclusions drawn from this research about patterns present in the larger medical device industry may be biased by practices specific to the regulation of robotic surgical devices. In particular, the technical complexity of robotic surgical devices may lead to a higher number of predicate devices than would be present in less-complex devices. The high number of predicates per generation in this investigation was determined to be partially responsible for the high rate of predicate creep, so other types of devices may not have such significant predicate creep. Moreover, given that for many years there was little or no competition to the Da Vinci Surgical Systems, we chose a device category for which there are likely to be fewer products available to cite as predicates than for products with higher levels of market competition.

Lastly, the device investigated in this study was cleared in 2009. In 2013, the FDA completed a small sample survey to better understand the causes behind adverse events in the Da Vinci Surgical system [21]. In 2014, the FDA took steps to change the review of robotically assisted surgical devices [22]. Thus, our review does not take the evolution of regulation that has happened since 2009. While this does not limit the use of this method to investigate predicate creep, it may impact how patterns are interpreted over time. It would be interesting to compare how the predicate network changes with newer models of the Da Vinci that may have been cleared under different processes and standards.

## Implications

**Implications for research.**   This investigation identified the presence of technological creep in predicate histories and made observations about common regulatory patterns and practices. However, the choice to focus this investigation on robotic surgical systems, a device type known for its technical complexity, may not be generalizable to other medical devices. Exploring the predicate history of additional device types would allow for comparison of regulatory patterns across device types and validate the conclusions of this investigation.

It might also be interesting to compare predicate history with patent history. While the 510 (k) process purports to identify predicate devices based on substantial equivalence in both functionality and technology, the patent process in the US requires proof that an idea is new and novel to secure a patent. Examination of the patent literature for devices which appear in the ancestral equivalence tree constructed from the regulatory history might allow for

identification of new technological characteristics introduced in each device. Theoretically, larger technological "leaps" present in substantial equivalence trees should correspond to a stronger patent presence for a given device.

**Implications for policy.** Looking at the regulatory process from an outside perspective, it is clear that predicate creep exists inherent within the substantial equivalence process. When this technology creep occurs on a small scale, introducing a new technology feature or application which slightly alters the form or function of the device while preserving the overall function and technological characteristics of the predicate, there is little potential impact to the safety of the public. In fact, purposeful inclusion of small amounts of technology creep is necessary to allow for innovation and improvement in medical device design. However, the effects of predicate creep over time have allowed for the development and clearance of entirely new devices without undergoing the stricter PMA approval process. Because this snowballing effect is directly dependent on small-scale predicate creep, it may be difficult to address the problem without negatively impacting the ability of manufacturers to bring innovative devices to market.

There have been calls for the FDA to develop a comprehensive, easily accessible database of predicate relationships. A complete map, including product codes and regulatory classifications would reveal common patterns in predicate relationships. The method used in this paper could then be used to identify critical points where a new device may be significantly different from the closest predicate in technology or intended use. At these critical points the FDA can then require additional scientific evidence of the overall device function, such as a small clinical trial, to ensure that no safety flaws have been introduced in the device due to predicate creep. This type of map would also help identify devices predicated on recalled devices or devices with known regulatory problems. This would allow for the continued use of small-scale predicate creep for technical innovation, while mitigating the introduction of untested technical characteristics and potential safety flaws over time.

Another important issue identified through this research is not the presence of predicate creep over time, but rather the sudden introduction of devices with high levels of technical complexity into the market through the 510(k) process. Starting in 2014, the FDA took steps to change the review of robotically assisted surgical devices [21]. Our review does not take these changes into account. They have also made changes that try to address the presence of extreme technology creep, so-called "leap" devices, in the regulatory process by creating more stringent guidelines for identification of predicate devices. These new guidelines reject the use of split predicates, where a device identifies one predicate for intended use and a separate predicate for technological characteristics. Interestingly, most "leap" devices identified in this research were not cleared using split predicates, but rather multiple predicates composed of "step" devices specifically designed to advance the desired use case for a particular technology.

It is possible that companies could take advantage of the substantial equivalence process by creating "step" devices, which are submitted and cleared for the specific purpose of serving as a predicate for technical characteristics, rather than as a marketable medical device. These "step" devices serve as intermediate predicates to allow more innovative devices with larger technological "leaps" into the market. Although devices with larger technological leaps are not necessarily unsafe or ineffective, for example the Da Vinci has remained on the market for over 15 years without a major recall, they do may possess more potential risk due to the fast-paced introduction of less-understood technologies into the marketplace. Although the FDA makes efforts to mitigate this risk through existing regulatory mechanisms, the lack of clearly defined substantial equivalence requirements makes it difficult to determine whether measures taken for a particular device are sufficient.

In the future, the FDA could make efforts to identify the "leap" devices and create a more targeted approval process that addresses new questions raised by these technologies. Defining clear guidelines for the amount of scientific evidence required based on the significance of new technological characteristics for device clearance would help reduce this inherent systematic risk.

For example, there may be different evidentiary requirements for a new device that is identical to an existing device but being used for a new application vs. a device utilizing a novel combination of technological characteristics from multiple previously cleared devices vs. a device that allows for a novel and significantly different use scenario.

## Conclusions

The primary method used to address these objectives was the development of multiple predicate ancestry trees using information available through FDA databases. Although the amount of available data is significantly limited due to missing data in the FDA database, the predicate traces developed contained enough information to determine if there was evidence of predicate creep. Through analysis of these traces, with a focus on regulatory information such as product classification codes, we were able to conclude that there is indeed both large and small scale predicate creep present within the 510(k) clearance process. It is important to note that while our research focuses on ways to identify predicate creep, it does not take the next step in identifying if this creep is, in fact, problematic. If all the preceding predicate devices are safe, and precautions are taken to mitigate small-scale predicate creep, in many cases devices exhibiting large-scale predicate creep may still be safe. Additional research is needed to assess the actual risk of technology creep across different types of medical devices and to develop mechanisms to identify when this creep may pose additional risk.

## Author Contributions

**Conceptualization:** Charlotte Lefkovich.

**Formal analysis:** Charlotte Lefkovich.

**Supervision:** Sandra Rothenberg.

**Writing – original draft:** Charlotte Lefkovich.

**Writing – review & editing:** Sandra Rothenberg.

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
