## [Decision Letter · Decision Letter 0]

14 Sep 2022

PONE-D-22-04558Identification of Predicate Creep under the 510K process: A Case Study of Robotic Surgical DevicesPLOS ONE

Dear Dr. Rothenberg,

Thank you for submitting your manuscript to PLOS ONE. After careful consideration, we feel that it has merit but does not fully meet PLOS ONE’s publication criteria as it currently stands. Therefore, we invite you to submit a revised version of the manuscript that addresses the points raised during the review process.

Please sufficiently address the comments and submit your revised manuscript in six months. If you will need more time than this to complete your revisions, please reply to this message or contact the journal office at plosone@plos.org. Please include the following items when submitting your revised manuscript:A rebuttal letter that responds to each point raised by the academic editor and reviewer(s). You should upload this letter as a separate file labeled 'Response to Reviewers'.A marked-up copy of your manuscript that highlights changes made to the original version. You should upload this as a separate file labeled 'Revised Manuscript with Track Changes'.An unmarked version of your revised paper without tracked changes. You should upload this as a separate file labeled 'Manuscript'.

We look forward to receiving your revised manuscript.

Kind regards,

Quanzeng Wang

Academic Editor

PLOS ONE

Journal Requirements:

3. We noted in your submission details that a portion of your manuscript may have been presented or published elsewhere. [this paper is from a Master's Thesis, which was published by Rochester Institute of Technology.] Please clarify whether this [conference proceeding or publication] was peer-reviewed and formally published. If this work was previously peer-reviewed and published, in the cover letter please provide the reason that this work does not constitute dual publication and should be included in the current manuscript.

5. We note that Figure 3 in your submission contain copyrighted images. All PLOS content is published under the Creative Commons Attribution License (CC BY 4.0), which means that the manuscript, images, and Supporting Information files will be freely available online, and any third party is permitted to access, download, copy, distribute, and use these materials in any way, even commercially, with proper attribution. For more information, see our copyright guidelines: http://journals.plos.org/plosone/s/licenses-and-copyright.

 a. You may seek permission from the original copyright holder of Figure 3 to publish the content specifically under the CC BY 4.0 license.

6. Please include your tables as part of your main manuscript and remove the individual files. Please note that supplementary tables (should remain/ be uploaded) as separate "supporting information" files.

Additional Editor Comments (if provided):

(1)

You cited the following paper in your manuscript.

B. M. Ardaugh, S. E. Graves, and R. F. Redberg, "The 510 (k) ancestry of a metal-on-metal hip implant," New England Journal of Medicine, vol. 368, no. 2, pp. 97-100, 2013.

This paper analyzed FDA documents obtained from both the agency’s database and Freedom of Information Office through the Freedom of Information Act (FOIA). In this manuscript, however, you only used data from the agency’s database, which is not complete. As a result, the predicate device trees (e.g., Fig. 3) are not accurate. Some devices were cleared after more generations than described in the manuscript. You should obtain a complete predicate history of Da Vinci Surgical system through FOIA and perform further analysis. If needed, you can contact the agency’s Division of Industry and Consumer Education (DICE, https://www.fda.gov/medical-devices/device-advice-comprehensive-regulatory-assistance/contact-us-division-industry-and-consumer-education-dice) for help to perform a through predicate analysis.

(2)

You mentioned that some devices were cleared without necessary clinical studies. However, you only analyzed data until 2008. Since the new 510(k) guidance published in 2014, FDA has already been steadily enhancing the review process of Robotic Assisted Surgery (RAS) devices to address risk. Clinical data has been required to clear new surgical procedures for Da Vinci Surgical system (e.g., K182371). You should discuss recent data from 2008 to present to provide a full and complete picture of FDA regulatory landscape.

(3)

If for whatever reason you cannot perform a thorough study as suggested, you should provide justifications and sufficiently discuss limitations of this study in the paper.

Reviewers' comments:

Reviewer's Responses to Questions

**Comments to the Author**

1. Is the manuscript technically sound, and do the data support the conclusions?

Reviewer #1: Partly

Reviewer #2: Partly

2. Has the statistical analysis been performed appropriately and rigorously? 

Reviewer #1: N/A

Reviewer #2: N/A

3. Have the authors made all data underlying the findings in their manuscript fully available?

Reviewer #1: Yes

Reviewer #2: Yes

4. Is the manuscript presented in an intelligible fashion and written in standard English?

Reviewer #1: Yes

Reviewer #2: Yes

5. Review Comments to the Author

Reviewer #1: 

The case of RASD is an unusual situation. There is likely not other case with greater creep. When the figures are separated from the text it makes if more challenging to read. There are other cases of creep for sure but this is not typical.

Reviewer #2: 

Thank you for the opportunity to review this important study on the role of predicate creep in medical device regulation. The excessive use of predicates, and the technological and regulatory divergence among devices in the Da Vinci lineage, is striking.

Several thematic and organizational comments on the manuscript below, with comments on Figures at the end:

– The literature already contains several reports about predicate creep, including a recent study focused specifically on some of the same Da Vinci robotic surgery systems that this study focuses on (https://pubmed.ncbi.nlm.nih.gov/35032697/). The studies have some differences; the one I have cited is more focused on ancestry, and this study is more focused on predicate creep. However, I think the authors would benefit from more clearly distinguishing their work and articulating what the new contribution to the literature is.

– While predicate creep is a tricky phenomenon to study, I am not entirely sure how the methodology of this study differs from previous research on predicate creep; as previous studies have also looked at predicates, decision memos, product codes, etc. The article may benefit from a more explicit discussion of what is new in their methodology (especially since the authors state on Page 17 that they “follow a method similar to Ardaugh and Zuckerman”). The findings can still be helpful even if the methodology is similar, but the manuscript’s narrative would need to be reframed.

– The discussion of using different Da Vinci databases to perform predicate traces is a bit confusing for the reader.

– One of the strongest findings to me is in Figures 5-7, which nicely capture the evolution in product codes and regulatory descriptions over time. This is a finding that certainly has important regulatory implications given that the FDA is moving to create product-specific guidance documents for 510(k).

– The introduction and literature review sections are lengthy and contain important information; however, given the length of the manuscript, these sections may benefit from revisions for brevity to improve the readability of the manuscript.

Specific stylistic comments:

– The manuscript itself is quite lengthy and would benefit from revisions for brevity.

– Throughout the document, the authors use the term “510K Process”. Per FDA documents and other published reports in the literature, I believe the more appropriate term is “510(k) pathway” rather than “process”.

– Throughout the document, the authors alternate between using “510K” and “510(k)”. Please revise to the latter, which is how the pathway is described in FDA documents.

– There are some typographical errors in the manuscript, and additional references may be needed at different junctures. Please see the comments below for specific instances.

The below comments are recommendations for specific-line edits. The manuscript version I was provided does not include line numbers; consequently, I have tried to provide page numbers (referring to the page in the document) and sentence quotes for the authors’ ease of reference:

– Page 8: in the sentence “…..drawn attention to this regulatory approval process”; suggest changing “approval” to either “clearance” or “authorization”. Devices cleared under 510(k) are not technically “approved”.

– Page 8: in the sentence “…predicate creep, a cycle of technology change”; according to the FDA’s definition of predicate creep, the changes may not just be limited to technology, but can also refer to indications. Would suggest revising to reflect this.

– Page 9: In the sentence starting “In the last decade…”, would suggest referencing the 2011 IOM report on 510(k), which references “creep” on page 89 and 230. https://nap.nationalacademies.org/download/13150 – Page 9: In the introduction, I suggest the authors explicitly reference the FDA’s 2009 report on the ReGen MenaFlex device, which (to my knowledge) is the first time the agency explicitly acknowledged the phenomenon of predicate creep and provided a formal definition: https://int.nyt.com/data/int-shared/nytdocs/docs/104/104.pdf.

– Page 9: In the sentence “…there is a lack of data to support concerns surrounding the 510(k)”; I am not sure this is true. There are many case studies in the literature (see Kadakia 2021 in JAMA IM, Freeman 2014 in Annals of Health Law, Ardaugh 2013 in NEJM) and some systematic studies as well (Pai 2021 in PLOS ONE, Zuckerman 2014 in JAMA IM). I believe the introduction might benefit from a more robust discussion of the existing literature, and how the methodology/intervention of this study contributes to and/or differs from existing approaches and knowledge.

– Page 10: In the sentence “…these methods will be tested on the Da Vinci Surgical System”; a recent study published in the International Journal of Surgery took this same approach (https://pubmed.ncbi.nlm.nih.gov/35032697/). I would suggest the authors contextualize their findings and methods given this recent research.

– Page 10: Rather than quoting the FDA definition, the authors could consider paraphrasing for brevity.

– Page 11: The sentence “…manufacturers are able to modify the wording of the stated intended use to make changes in device function appear minimal” may benefit from a citation given it is a claim of manufacturer intent.

– Page 11: In the sentence “In his concurring opinion…”; I believe this should be “her”, as Justice O’Connor is a female.

– Page 12: In the sentence “…deemed safe based on market performance”; would clarify for the reader what “market performance” actually means.

– Page 13: In the sentence “…maintained by the FDA, with mixed results”; I am not sure what makes the previous studies “mixed”. Please clarify.

– Page 13: In the sentence “…it is hard to do without the use of mandatory clinical trials, which would defeat the purpose of the 510(k) process entirely”. I am not sure this is true. The authors could consider the role of registries and post-market surveillance platforms in advancing evidence generation, and clearer premarket standards in mitigating predicate creep.

– Page 14: In the sentence “…documents obtained through the FDA 510(f) database”; there appears to be a typo, the “(f)” should be a “(k)”

– Page 14: The authors write “Neither article, however, clearly specified the exact methodology used to trace predicates”. I am not sure this is true, as the preceding sentence indicates that Zuckerman used the 510(k) database, and the original manuscript cited here states in their methods section that “we analyzed the type of scientific evidence the company provided to the FDA and the public to support the claim of substantial equivalence to a device already on the market or to establish safety or effectiveness” – which appears to be a similar method to what the authors are proposing? The main difference appears to be that the previous study did not look at differences in product code. (Additionally, the authors state on Page 17 that they “follow a method similar to Ardaugh and Zuckerman” – so it’s not clear what the flaw/differences are).

– Page 15: The sentence “…more precise control and motion” is a claim of performance that should include a citation. Similarly, the argument in the following sentence about “…Intuitive’s strong patent foothold” also requires a citation.

– Page 15: The paragraph “The Da Vinci is an interesting case study” may benefit from being moved to the methods. It should also include a discussion of a recent paper published on its 510(k) ancestry in the International Journal of Surgery.

– Page 16: In the sentence “For this paper, most data….”; please define what the “data” are.

– Page 16: In the sentence, “…were collected through 510(k) approval database”; a “the” appears to be missing, and “approval” should be “clearance”.

– Page 16: In the sentence “When either of these databases did not provide the needed information, we also conducted targeted search…”; the “either” should be “neither”, and an “a” is missing before “targeted”.

–Page 17: Portions of the figure description could be moved from the methods in to a caption for the figure to improve brevity.

–Page 18: In the sentence “….increased number of predicates appear make traditional tree diagrams unwieldy”; a “to” appears to be missing”.

– Page 18: The majority of the paragraph under the header, “Measuring Predicate Creep” could be deleted as this set-up has already been completed in the Introduction/Literature Review (which is the more appropriate place for it).

– Page 21: The sentence “Since the successful performance of each predicate device in the market is part of the body of evidence to support the safety claims of the new device, a smaller number of unique devices with market performance data effectively reduces the level of assurance of safety for the subject device” – does not make sense. Per the IOM report and the authors’ own arguments in the introduction, the 510(k) pathway does not support justifications of safety and effectiveness. Additionally, per the authors’ arguments, shouldn’t the quality of the substantial equivalence determination rather than the number of unique devices be the real determinant of safety?

– Page 21: In the section on “Direct Comparison of Technological Characteristics”; it is not clear what makes “the degree of technological differences ‘striking’” or (per page 22) what constitutes a “large change in technology within a single predicate relationship”. (Some of this is of course intuitive, but it is worth spelling out for the reader – because by the FDA’s guidance, the subject and predicate are not required to be identical, so some differences are expected”). For example, the authors could consider interweaving a discussion of the clinical/non-clinical evidence here.

– Page 24: What is the difference between “device instances” and “unique devices”?

– Page 24: “Grouping by regulatory mechanisms”; devices can have their product codes reclassified over time; was that the case for any of the devices in the sample?

– Page 29: The sentence “…methods of predicate analysis; The” I think the semicolon is supposed to be a period.

– Page 30: In the sentence “the 510(k) process was officially implemented via the 1990 Safe Medical Devices Act” – technically the pathway was established as part of the 1976 device amendments. Additionally, this paragraph could be potentially condensed, as the use of these different forms is a moot point since digital copies anyway are not available until ~2000.

– Page 32: The first paragraph under “Implications for Research” should be moved to the limitations section (as it is essentially a question of generalizability)

– Page 34: The sentence “Although devices with larger…” is grammatically off (the clause in the middle about Da Vinci creates an awkward break”).

– Page 35: The sentence “The FDA has recently begun taking steps…” could use a citation. Also, I am not sure that “split predicates” are really the problem for “leap devices” as identified in the authors’ research.

Comments on Figures

– Figure 1: While this figure is well-intentioned, I don’t think the message is quite clear to the reader. Perhaps some simple labeling like “Device 1, Device 2”, etc. could be helpful. Also perhaps worth noting somewhere here the lack of testing.

– Figure 2: Would recommend reformatting this figure, given the unevenness of the red branch line. Could either draw a more symmetric shape, or could consider just highlighting the lines in colors rather than creating a box.

– Figures 5, 6, 7: These figures are really nicely done, and nicely capture the regulatory effect of predicate creep. Kudos to the authors.

6. PLOS authors have the option to publish the peer review history of their article (what does this mean?). If published, this will include your full peer review and any attached files.

Reviewer #1: No

Reviewer #2: No

---

## [Author Response · Author response to Decision Letter 0]

23 Dec 2022

Thank you for giving us the opportunity to revise this paper. Based on the reviewer comments we have significantly shortened and altered the framing of this paper. We now focus on what reviewer 2 thought was the primary contribution of the paper, identification of potential “predicate creep” through the use of product codes and regulatory classifications. Below you will find a detailed response to yours and the reviewers’ comments. If you refer to the attached response documents, our responses are in italics.

We have reformatted the title page and paper according to these guidelines.

We have included the title page.

3. We noted in your submission details that a portion of your manuscript may have been presented or published elsewhere. [this paper is from a Master's Thesis, which was published by Rochester Institute of Technology.] Please clarify whether this [conference proceeding or publication] was peer-reviewed and formally published. If this work was previously peer-reviewed and published, in the cover letter please provide the reason that this work does not constitute dual publication and should be included in the current manuscript.

We have included this explanation in the cover letter.

We have made a short caption for each figure in the manuscript.

5. We note that Figure 3 in your submission contain copyrighted images. All PLOS content is published under the Creative Commons Attribution License (CC BY 4.0), which means that the manuscript, images, and Supporting Information files will be freely available online, and any third party is permitted to access, download, copy, distribute, and use these materials in any way, even commercially, with proper attribution. For more information, see our copyright guidelines: http://journals.plos.org/plosone/s/licenses-and-copyright.

 a. You may seek permission from the original copyright holder of Figure 3 to publish the content specifically under the CC BY 4.0 license.

This figure has been removed from the paper.

6. Please include your tables as part of your main manuscript and remove the individual files. Please note that supplementary tables (should remain/ be uploaded) as separate "supporting information" files.

Additional Editor Comments (if provided):

(1)

You cited the following paper in your manuscript.

B. M. Ardaugh, S. E. Graves, and R. F. Redberg, "The 510 (k) ancestry of a metal-on-metal hip implant," New England Journal of Medicine, vol. 368, no. 2, pp. 97-100, 2013.

This paper analyzed FDA documents obtained from both the agency’s database and Freedom of Information Office through the Freedom of Information Act (FOIA). In this manuscript, however, you only used data from the agency’s database, which is not complete. As a result, the predicate device trees (e.g., Fig. 3) are not accurate. Some devices were cleared after more generations than described in the manuscript. You should obtain a complete predicate history of Da Vinci Surgical system through FOIA and perform further analysis. If needed, you can contact the agency’s Division of Industry and Consumer Education (DICE, https://www.fda.gov/medical-devices/device-advice-comprehensive-regulatory-assistance/contact-us-division-industry-and-consumer-education-dice) for help to perform a through predicate analysis.

We did consider requesting additional data through FOIA. We decided against this route, however, since earlier experience with the FOIA process resulted in response times of over 6 months. Moreover, once the first level of predicates are requested, we would then need to see what predicates are used for any new devices in the tree, which would then take another 6 months. We did try sending some requests directly to the companies, but these requests were uniformly unanswered. Given we are looking backwards in time, the fact that some predicates are missing actually makes any finding of predicate creep more significant. Additional predicate data would either show no change in the amount of predicate creep or would show increased predicate creep. We now include the following in our paper:

“This study was limited by several factors. First, the investigations performed in this study were based solely on data available publicly through FDA databases. Predicate devices are identified within the database only when application summaries are provided, which was not required for inclusion until the mid-2000’s. This significantly limits the number of devices with traceable predicate histories. Moreover, the level of information contained within these summaries varies widely, and in many cases these summaries do not exist at all, due to the gradual evolution of requirements for 510(k) application submissions. We did consider requesting additional data through FOIA, decided against this route given that it would add a significant amount of time to the data gathering and there was a precedent in prior research to rely primarily on information provided in the FDA website [11]. Additionally, given we are looking backwards in time, the fact that some predicates are missing actually makes any finding of predicate creep more significant.” 

(2)

You mentioned that some devices were cleared without necessary clinical studies. However, you only analyzed data until 2008. Since the new 510(k) guidance published in 2014, FDA has already been steadily enhancing the review process of Robotic Assisted Surgery (RAS) devices to address risk. Clinical data has been required to clear new surgical procedures for Da Vinci Surgical system (e.g., K182371). You should discuss recent data from 2008 to present to provide a full and complete picture of FDA regulatory landscape.

Since we are focus on one particular device and looking back, we did not feel the need to include more recent data. Given the refocusing of the paper on the method identifying predicate creep reviewer 2 thought was the primary contribution of the paper, identification of potential “predicate creep” through the use of product codes and regulatory classifications, providing more recent data does not add to the goal of the analysis. You are correct that we make statements about the FDA process without including more recent changes in process. We have addressed this in the limitations.

We now state:

“Lastly, the device investigated in this study was cleared in 2009. In 2013, the FDA completed a small sample survey to better understand the causes behind adverse events in the da Vinci Surgical system [20]. In 2014, the FDA took steps to change the review of robotically assisted surgical devices [21]. Thus, our review does not take the evolution of regulation that has happened since 2009. While this does not limit the use of this method to investigate predicate creep, it may impact how patterns are interpreted over time. It would be interesting to compare how the predicate network changes with newer models of the da Vinci that may have been cleared under different processes and standards.”

(3)

If for whatever reason you cannot perform a thorough study as suggested, you should provide justifications and sufficiently discuss limitations of this study in the paper.

Please see the additions to the limitation as listed above.

Comments to the Author

Review Comments to the Author

Reviewer #1: 

The case of RASD is an unusual situation. There is likely not other case with greater creep. When the figures are separated from the text it makes if more challenging to read. There are other cases of creep for sure but this is not typical.

Thank you for your comment. We agree that this level of creep may not be seem across other medical devices. It is, however, a good case to review ways to measure predicate creep. We discuss the issue of generalizability as a limitation of the study. We apologize for the separation of the figures and also find this more difficult to read. Unfortunately, that was the format that was requested by the journal.

Reviewer #2: 

Thank you for the opportunity to review this important study on the role of predicate creep in medical device regulation. The excessive use of predicates, and the technological and regulatory divergence among devices in the Da Vinci lineage, is striking.

Thank you in advance for these helpful comments. You are clearly very familiar with this literature and the FDA processes. Your comments have led us to new, important, references. Perhaps more important, it has led to focus the paper on the part of the analysis you found most interesting, which was the second half the paper. By focusing on this part of the paper, the paper is hopefully more concise and makes a clearer contribution.

Several thematic and organizational comments on the manuscript below, with comments on Figures at the end:

– The literature already contains several reports about predicate creep, including a recent study focused specifically on some of the same Da Vinci robotic surgery systems that this study focuses on (https://pubmed.ncbi.nlm.nih.gov/35032697/). The studies have some differences; the one I have cited is more focused on ancestry, and this study is more focused on predicate creep. However, I think the authors would benefit from more clearly distinguishing their work and articulating what the new contribution to the literature is.

Thank you for pointing out this article. We have integrated it into our paper. In addition, as summarized in response to this and your next comment. We agree that we don’t sufficiently differentiate our paper. We have modified the end of the introduction in a way that both highlights the contributions of past research, and differentiates the contribution of our own paper. Please see the changes, placed after your next comment, to see the new framing.

– While predicate creep is a tricky phenomenon to study, I am not entirely sure how the methodology of this study differs from previous research on predicate creep; as previous studies have also looked at predicates, decision memos, product codes, etc. The article may benefit from a more explicit discussion of what is new in their methodology (especially since the authors state on Page 17 that they “follow a method similar to Ardaugh and Zuckerman”). The findings can still be helpful even if the methodology is similar, but the manuscript’s narrative would need to be reframed.

We completely agree with this assessment that the technical comparison method is essentially the same as the one just by Ardaugh et. al (2013). This has led us to realize that our most important contribution is essentially our second method, which focuses on regulatory classification. The benefit of this approach is that, once the data is entered, can be done fairly quickly and be used to identify areas in a predicate network that one can focus on for direct technical comparisons. We have changed the paper narrative to this effect and have added the following:

“Within the 510(k) process, requirements for non-clinical testing are used to mitigate the risk associated with small scale predicate creep, which in most cases works effectively. However, past research on predicate networks has highlighted the lack of publicly available non-clinical scientific data in substantial equivalence determinations. Both Zuckerman et al. [11] and Liebeskind et al. [15] used the FDA 510(k) Database and/or FOIA requests to trace the predicate history of different medical devices and found that there was a dearth of publicly available scientific data to support the claim of substantial equivalence. Liebeskind et al. [15], in particular, traced the predicate ancestry of robotic surgical systems, and found that (92.7%) 510(k) clearances did not submit clinical data, including 73 (27.9%) that did not submit any supporting data. These studies suggest that non-clinical evidence to support substantial equivalence claims may be insufficient to ensure safety when there is predicate creep.

Past research on predicate creep primarily involves creating predicate ancestry trees and using direct technical comparisons of devices and their predicates. Such studies have identified concerning predicate creep in surgical meshes [6] and Pathwork Tissue of Origin Test [9], DePuy ASR XL Acetabular Cup System [5], power morcellators [16], among others. Ardaugh et al. [5], for example, studied the metal-on-metal hip implant over five decades with the purpose of identifying the cause of safety flaws present in the design. They identify that a unique combination of three characteristics in the ASR XL were approved though substantial equivalence of “split predicates,” where you compare characteristic to different predicate devices. Since all three devices were deemed safe based on clinical trials and safe use of predicates in the consumer market, and the XL simply combined parts of the devices, it was placed on the market without undergoing clinical testing and resulting product had to be recalled for particle shedding [5]. 

Direct technical comparisons can be challenging due to a lack of information, as discussed above, or the cumbersome size of predicate ancestry trees, particularly for newer devices. Using FOIA requests to attain more detailed information can also be onerous. This paper proposes another way to identify potentially important instances of predicate creep in a large predicate network through analysis of product classes and regulatory categorizations. We test this method on a well-known medical device, the Intuitive Surgical da Vinci Si robotic surgical system.”

– The discussion of using different Da Vinci databases to perform predicate traces is a bit confusing for the reader.

– One of the strongest findings to me is in Figures 5-7, which nicely capture the evolution in product codes and regulatory descriptions over time. This is a finding that certainly has important regulatory implications given that the FDA is moving to create product-specific guidance documents for 510(k).

We agree that this should be more of the focus of the paper, and is the main finding. This has been made clearer in the introduction and the conclusion.

– The introduction and literature review sections are lengthy and contain important information; however, given the length of the manuscript, these sections may benefit from revisions for brevity to improve the readability of the manuscript.

Specific stylistic comments:

– The manuscript itself is quite lengthy and would benefit from revisions for brevity.

We have significantly shortened the paper by reducing the literature review and taking out the material on the first Da Vinci technical comparison. 

– Throughout the document, the authors use the term “510K Process”. Per FDA documents and other published reports in the literature, I believe the more appropriate term is “510(k) pathway” rather than “process”.

We have completed this change.

– Throughout the document, the authors alternate between using “510K” and “510(k)”. Please revise to the latter, which is how the pathway is described in FDA documents.

We have completed this change.

– There are some typographical errors in the manuscript, and additional references may be needed at different junctures. Please see the comments below for specific instances.

The below comments are recommendations for specific-line edits. The manuscript version I was provided does not include line numbers; consequently, I have tried to provide page numbers (referring to the page in the document) and sentence quotes for the authors’ ease of reference:

– Page 8: in the sentence “…..drawn attention to this regulatory approval process”; suggest changing “approval” to either “clearance” or “authorization”. Devices cleared under 510(k) are not technically “approved”.

Thank you for catching this. Since the FDA website refers to this is clearance – we have changed this to the term “clearance” throughout the paper

– Page 8: in the sentence “…predicate creep, a cycle of technology change”; according to the FDA’s definition of predicate creep, the changes may not just be limited to technology, but can also refer to indications. Would suggest revising to reflect this.

Thank you – using the reference provided below on the ReGen MenaFlex, we have changed the sentence to the following:

“One of the specific concerns in the 510(k) process is the risk of predicate creep, a cycle of technology change through repeated clearance of devices based on predicates with slightly different technological characteristics, such as materials and power sources, or have indications for different anatomical sites [3-4].”

– Page 9: In the sentence starting “In the last decade…”, would suggest referencing the 2011 IOM report on 510(k), which references “creep” on page 89 and 230. https://nap.nationalacademies.org/download/13150 – 

Thank you for this reference. There was quite a bit of valuable material in it and we now reference it.

Page 9: In the introduction, I suggest the authors explicitly reference the FDA’s 2009 report on the ReGen MenaFlex device, which (to my knowledge) is the first time the agency explicitly acknowledged the phenomenon of predicate creep and provided a formal definition: https://int.nyt.com/data/int-shared/nytdocs/docs/104/104.pdf.

Thank you for this suggestion. We have reviewed this report and now reference it in the paper.

– Page 9: In the sentence “…there is a lack of data to support concerns surrounding the 510(k)”; I am not sure this is true. There are many case studies in the literature (see Kadakia 2021 in JAMA IM, Freeman 2014 in Annals of Health Law, Ardaugh 2013 in NEJM) and some systematic studies as well (Pai 2021 in PLOS ONE, Zuckerman 2014 in JAMA IM). I believe the introduction might benefit from a more robust discussion of the existing literature, and how the methodology/intervention of this study contributes to and/or differs from existing approaches and knowledge.

We completely agree. In fact, we contradict this statement in the literature review. We now have changed this part of the introduction to read as follows and document some of the specific problems in the literature review section of the paper:

“In the last decade, several high-profile device recalls have drawn attention to this regulatory clearance process and researchers have raised concerns about the validity of the 510(k) process as a broad clearance mechanism. These concerns include, but are not limited to, a lack of a clear definition of “intended use,” using recalled devices as predicates, lack of publicly available data regarding predicates and determinations of substantial equivelence, equating substantial equivelence to safety, and a lack of sufficient non-clinical data to support claims of substantial equivelence, among others. As outlined in the literature review below, there is a body of research that documents these problems and the resulting safety issues that have stemmed from them.” 

– Page 10: In the sentence “…these methods will be tested on the Da Vinci Surgical System”; a recent study published in the International Journal of Surgery took this same approach (https://pubmed.ncbi.nlm.nih.gov/35032697/). I would suggest the authors contextualize their findings and methods given this recent research.

Again, thank you for identifying this article. We now highlight this article and its findings, but also state how what they do is different than what we will be doing. We have added the following:

“Within the 510(k) process, requirements for non-clinical testing are used to mitigate the risk associated with small scale predicate creep, which in most cases works effectively. However, past research on predicate networks has highlighted the lack of publicly available non-clinical scientific data in substantial equivalence determinations. Both Zuckerman et al. [11] and Liebeskind et al. [15] used the FDA 510(k) Database and/or FOIA requests to trace the predicate history of different medical devices and found that there was a dearth of publicly available scientific data to support the claim of substantial equivalence. Liebeskind et al. [15], in particular, traced the predicate ancestry of robotic surgical systems, and found that (92.7%) 510(k) clearances did not submit clinical data, including 73 (27.9%) that did not submit any supporting data. These studies suggest that non-clinical evidence to support substantial equivalence claims may be insufficient to ensure safety when there is predicate creep.

Past research on predicate creep primarily involves creating predicate ancestry trees and using direct technical comparisons of devices and their predicates. Such studies have identified concerning predicate creep in surgical meshes [6] and Pathwork Tissue of Origin Test [9], DePuy ASR XL Acetabular Cup System [5], power morcellators [16], among others. Ardaugh et al. [5], for example, studied the metal-on-metal hip implant over five decades with the purpose of identifying the cause of safety flaws present in the design. They identify that a unique combination of three characteristics in the ASR XL were approved though substantial equivalence of “split predicates,” where you compare characteristic to different predicate devices. Since all three devices were deemed safe based on clinical trials and safe use of predicates in the consumer market, and the XL simply combined parts of the devices, it was placed on the market without undergoing clinical testing and resulting product had to be recalled for particle shedding [5]. 

Direct technical comparisons can be challenging due to a lack of information, as discussed above, or the cumbersome size of predicate ancestry trees, particularly for newer devices. Using FOIA requests to attain more detailed information can also be onerous. This paper proposes another way to identify potentially important instances of predicate creep in a large predicate network through analysis of product classes and regulatory categorizations. We test this method on a well-known medical device, the Intuitive Surgical da Vinci Si robotic surgical system.”

– Page 10: Rather than quoting the FDA definition, the authors could consider paraphrasing for brevity.

We have removed this long quote, as we feel it was not needed.

– Page 11: The sentence “…manufacturers are able to modify the wording of the stated intended use to make changes in device function appear minimal” may benefit from a citation given it is a claim of manufacturer intent.

This is a good observation. Rereading the cited papers, the issue of manufacturers intent was referred to papers that focused on the process in the NICE Evaluation Pathway (EP) Programme in the United Kingdom. Thus, we have changed the sentence to be as follows, which more accurately reflects the sentiment in the studies focused on the 510(k) process:

“Many of the critiques of the 510(k) process relate to the notion of substantial equivelence. First, due to the lack of a clear official definition for the key terms “indications for use” and “intended use”, the FDA has allowed permissive interpretation of these terms by applicants and inconsistent use of the terms across reviewers [7-8]. Over time, this has resulted in the clearance of significantly altered devices, or even novel devices, as substantially equivalent to established predicates [7-9].”

– Page 11: In the sentence “In his concurring opinion…”; I believe this should be “her”, as Justice O’Connor is a female.

Great catch. We actually took out this sentence to make the review more concise.

– Page 12: In the sentence “…deemed safe based on market performance”; would clarify for the reader what “market performance” actually means.

We have changed the sentence to read as follows:

“Since all three devices were deemed safe based on clinical trials and safe use of predicates in the consumer market, and the XL simply combined parts of the devices, it was placed on the market without undergoing clinical testing and resulting product had to be recalled for particle shedding [5]. “

– Page 13: In the sentence “…maintained by the FDA, with mixed results”; I am not sure what makes the previous studies “mixed”. Please clarify.

We have removed this sentence.

– Page 13: In the sentence “…it is hard to do without the use of mandatory clinical trials, which would defeat the purpose of the 510(k) process entirely”. I am not sure this is true. The authors could consider the role of registries and post-market surveillance platforms in advancing evidence generation, and clearer premarket standards in mitigating predicate creep.

We ended up removing this sentence, but you are correct. As the editor has indicated, there has been attention by the FDA to both of these areas for robotic surgical devices. Documents describing these changes (or intended changes) are now referenced in the limitations sections. 

– Page 14: In the sentence “…documents obtained through the FDA 510(f) database”; there appears to be a typo, the “(f)” should be a “(k)”

This has been fixed.

– Page 14: The authors write “Neither article, however, clearly specified the exact methodology used to trace predicates”. I am not sure this is true, as the preceding sentence indicates that Zuckerman used the 510(k) database, and the original manuscript cited here states in their methods section that “we analyzed the type of scientific evidence the company provided to the FDA and the public to support the claim of substantial equivalence to a device already on the market or to establish safety or effectiveness” – which appears to be a similar method to what the authors are proposing? The main difference appears to be that the previous study did not look at differences in product code. (Additionally, the authors state on Page 17 that they “follow a method similar to Ardaugh and Zuckerman” – so it’s not clear what the flaw/differences are).

We have removed this statement, and more clearly differentiate our method from Zuckerman and others as focusing on product codes and regulatory classifications.

– Page 15: The sentence “…more precise control and motion” is a claim of performance that should include a citation. Similarly, the argument in the following sentence about “…Intuitive’s strong patent foothold” also requires a citation.

We have added recent references to support these two statements.

“The Da Vinci Surgical System is a robotic-based laparoscopic surgical tool initially approved by the FDA in 2000 which replaces a surgeon’s hands with robotic arms for more precise control and motion [17]. While Intuitive has subsequently brought multiple iterations of the Da Vinci to market, it remains the only full RAS platform on the market as competitors struggle to develop a viable competitor around Intuitive’s strong patent foothold [18].”

– Page 15: The paragraph “The Da Vinci is an interesting case study” may benefit from being moved to the methods. It should also include a discussion of a recent paper published on its 510(k) ancestry in the International Journal of Surgery.

We have made both of these changes. In the methods we state:

“The Da Vinci Surgical System is a robotic-based laparoscopic surgical tool initially approved by the FDA in 2000 which replaces a surgeon’s hands with robotic arms for more precise control and motion [17]. While Intuitive has subsequently brought multiple iterations of the Da Vinci to market, it remains the only full RAS platform on the market as competitors struggle to develop a viable competitor around Intuitive’s strong patent foothold [18]. The Da Vinci itself was initially approved under a 510(k) application based on a complex web of component-level substantial equivalence, most likely supplemented by additional testing. The Da Vinci was chosen because its function is well documented. Moreover, there is a non-negligible number of malfunctions that have occurred during the use of these types of devices [19].”

– Page 16: In the sentence “For this paper, most data….”; please define what the “data” are.

We have changed this sentence to “For this paper information on the devices within the Da Vinci Si Model predicate network”

– Page 16: In the sentence, “…were collected through 510(k) approval database”; a “the” appears to be missing, and “approval” should be “clearance”.

We have changed this sentence to read: through the 510(k) clearance database

– Page 16: In the sentence “When either of these databases did not provide the needed information, we also conducted targeted search…”; the “either” should be “neither”, and an “a” is missing before “targeted”.

In shortening the paper, we deleted this particular sentence. It now reads “Information on the devices within the Da Vinci Si Model predicate network collected through the 510(k) clearance database. We also searched other data bases using the search function on the FDA web site, which returns results from all FDA publications, including database information, conference presentations, regulations, and internal memos.”

–Page 17: Portions of the figure description could be moved from the methods in to a caption for the figure to improve brevity.

–Page 18: In the sentence “….increased number of predicates appear make traditional tree diagrams unwieldy”; a “to” appears to be missing”.

We have changed the sentence: “The numbers of predicates in the Da Vinci Si predicate ancestry network made traditional tree diagrams too unwieldy, so an alternate diagram structure known as a network map was used to display predicate relationships within the clearance ancestry.”

– Page 18: The majority of the paragraph under the header, “Measuring Predicate Creep” could be deleted as this set-up has already been completed in the Introduction/Literature Review (which is the more appropriate place for it).

This method description has been condensed (and some of it deleted given the paper refocusing).

– Page 21: The sentence “Since the successful performance of each predicate device in the market is part of the body of evidence to support the safety claims of the new device, a smaller number of unique devices with market performance data effectively reduces the level of assurance of safety for the subject device” – does not make sense. Per the IOM report and the authors’ own arguments in the introduction, the 510(k) pathway does not support justifications of safety and effectiveness. Additionally, per the authors’ arguments, shouldn’t the quality of the substantial equivalence determination rather than the number of unique devices be the real determinant of safety?

In shortening the paper, this sentence was removed. However, it does raise an interesting question that could be explored empirically. 

– Page 21: In the section on “Direct Comparison of Technological Characteristics”; it is not clear what makes “the degree of technological differences ‘striking’” or (per page 22) what constitutes a “large change in technology within a single predicate relationship”. (Some of this is of course intuitive, but it is worth spelling out for the reader – because by the FDA’s guidance, the subject and predicate are not required to be identical, so some differences are expected”). For example, the authors could consider interweaving a discussion of the clinical/non-clinical evidence here.

We have removed this section.

– Page 24: What is the difference between “device instances” and “unique devices”?

This sentence has been reworded for clarity “This network map included 2618 device instances (i.e. a device is cited as a predicate), with a total of 50 unique devices.”

– Page 24: “Grouping by regulatory mechanisms”; devices can have their product codes reclassified over time; was that the case for any of the devices in the sample?

Yes, there were some products for which product codes were changed. These are listed in the table as having multiple product codes. For example, Guidant Microwave Ablation System was first coded as NAY and then given a subsequent product code OCL. For this particular device, which was cleared by the FDA in 2002, a letter notifying the manufacturer of this change was sent in 2008. Not all devices with multiple codes, however, had similar documentation as to when the code was switched. 

– Page 29: The sentence “…methods of predicate analysis; The” I think the semicolon is supposed to be a period.

This sentence was changed.

– Page 30: In the sentence “the 510(k) process was officially implemented via the 1990 Safe Medical Devices Act” – technically the pathway was established as part of the 1976 device amendments. Additionally, this paragraph could be potentially condensed, as the use of these different forms is a moot point since digital copies anyway are not available until ~2000.

We now simply state: Due to changes in regulation over time, the level of information available in the 510(k) database varied widely based on the date of approval.

– Page 32: The first paragraph under “Implications for Research” should be moved to the limitations section (as it is essentially a question of generalizability)

You are correct – it has been moved and merged into the limitation section

– Page 34: The sentence “Although devices with larger…” is grammatically off (the clause in the middle about Da Vinci creates an awkward break”).

We have changed this to read: “Devices with larger technological leaps are not necessarily unsafe or ineffective. For example, the Da Vinci has remained on the market for over 15 years without a major recall. Devices with large technological leaps, however, may possess more potential risk due to the fast-paced introduction of less-understood technologies into the marketplace.”

– Page 35: The sentence “The FDA has recently begun taking steps…” could use a citation. Also, I am not sure that “split predicates” are really the problem for “leap devices” as identified in the authors’ research.

You are absolutely correct that this paragraph was not well cited and was internally inconsistent. We ended up deleting this paragraph.

Comments on Figures

– Figure 1: While this figure is well-intentioned, I don’t think the message is quite clear to the reader. Perhaps some simple labeling like “Device 1, Device 2”, etc. could be helpful. Also perhaps worth noting somewhere here the lack of testing.

We have deleted this figure with the new structure of the paper. 

– Figure 2: Would recommend reformatting this figure, given the unevenness of the red branch line. Could either draw a more symmetric shape, or could consider just highlighting the lines in colors rather than creating a box.

We have deleted this figure with the new structure of the paper.

– Figures 5, 6, 7: These figures are really nicely done, and nicely capture the regulatory effect of predicate creep. Kudos to the authors.

Thank you. We intend to be the focus of the revised paper.

---

## [Decision Letter · Decision Letter 1]

23 Jan 2023

PONE-D-22-04558R1Identification of predicate creep under the 510(k) process: A case study of a robotic surgical devicePLOS ONE

Dear Dr. Rothenberg,

Thank you for submitting your manuscript to PLOS ONE. After careful consideration, we feel that it has merit but does not fully meet PLOS ONE’s publication criteria as it currently stands. Therefore, we invite you to submit a revised version of the manuscript that addresses the points raised during the review process.

We look forward to receiving your revised manuscript.

Kind regards,

Quanzeng Wang

Academic Editor

PLOS ONE

Journal Requirements:

Reviewers' comments:

Reviewer's Responses to Questions

**Comments to the Author**

1. If the authors have adequately addressed your comments raised in a previous round of review and you feel that this manuscript is now acceptable for publication, you may indicate that here to bypass the “Comments to the Author” section, enter your conflict of interest statement in the “Confidential to Editor” section, and submit your "Accept" recommendation.

Reviewer #2: All comments have been addressed

2. Is the manuscript technically sound, and do the data support the conclusions?

Reviewer #2: Yes

3. Has the statistical analysis been performed appropriately and rigorously? 

Reviewer #2: N/A

4. Have the authors made all data underlying the findings in their manuscript fully available?

Reviewer #2: Yes

5. Is the manuscript presented in an intelligible fashion and written in standard English?

Reviewer #2: Yes

6. Review Comments to the Author

Reviewer #2: Thank you very much for the opportunity to review a revised version of the manuscript by Lefkovich and Rothenberg. The authors should be applauded for their detailed and thorough revisions of the manuscript. I have included below some further substantive and stylistic comments for their consideration. Please note that page numbers below correspond to the 103-page document I was sent for my review, which includes both the authors’ responses, the clean version of the manuscript, and the tracked changes version of the manuscript.

Substantive:

-The updated focus on product codes is nicely done, and a good addition to the literature. The revised Figure 2 captures this quite well in particular. Kudos.

-In the discussion, I think the authors should be clear that the limitations of their work are not simply to a specific type of device (Robotic Surgery devices), but also to a device category which is highly consolidated (limited competition – which means there are fewer available predicates that likely can be cited) and is by definition composed of multiple different devices (a similar issue for devices like orthopedic implants, but perhaps less of an issue for devices such as respiratory machines or endovascular catheters)

-Introduction (page 18): the paragraph starting with “FDA has responded to these concerns…” could be deleted from the introduction. Some of this content is already in the discussion, and that is a more appropriate place for it. This change would also help shorten the introduction.

-Literature Review (page 23): Please revise this sentence: “The uniqueness of the device as an emerging technology makes it representative of the many challenges…”. It does not follow that a “unique” example can also be “representative”.

-When tracking product code divergence, did the authors also confirm that the change in product code was a decision by the manufacture, and not because of a reclassification order by FDA? As the latter would mean that it wasn’t so much an instance of “predicate creep” as it was updated regulation

-Data Analysis (page 29): “This illustrates how larger jumps in the technological complexity of devices new devices can occur through the 510(k) process” – great summary statement. Note that there seems to be a typo, with “devices” repeated twice. Otherwise, this well captures the contributions of this study.

-Discussion (page 32): The sentence “…to reduce barriers to bringing new medical devices” is not quite correct (that is perhaps a summation of the “Least Burdensome Principle”). The purpose of 510(k) is to facilitate incremental innovation in medical device development; not just reduce barriers. Please correct.

-Discussion (page 32): The sentence “..predicate creep where the new device can be guaranteed safe…”. “Guarantee” may not be appropriate here. The IOM report on 510(k) is clear that 510(k) clearance does not demonstrate safety. Would consider revising.

-Discussion (page 33): “We were also able to see the absorption of a primary predicate device function into a secondary system function” – a really important interesting insight. Nicely done.

-Limitations (page 35): In the sentence “scope of this research to robotic surgical devices”; it also may be useful to point about the fact about consolidation/monopolization of this space, which makes it harder to generalize to other product codes where there may be more device makers (and hence, more predicates)

-Implications for Research (Page 36): the point about patents is really interesting. Of note, there’s no good resource for this for device (e.g., there’s no equivalent of the FDA Orange Book for generic medicines as there is for devices).

-Implications for Policy (Page 38): The discussion about Step and Leap devices is interesting; I think the authors could provide more weight and clarity to this section by inverting the order of the current organization. First, highlight what FDA has already done, and then offer recommendations; bounces around a little too much right now.

Stylistic

-The figures appear a bit blurry in my version of the PDF. Assuming Editors will work with the authors to ensure the figures in the final version of the paper are of a higher resolution

-At several points in the paper, the authors use the word “approved” (for example, page 21 of the Literature “These predicates can have also been approved…”). 510(k) does not provide “approvals”; the appropriate term is either “clearance” or “authorization”. Please revise all references throughout

-Throughout the article, the authors alternate from “Da Vinci” and “Da Vinci Si”. Please use just one consistent name.

-In the Implications for Future Research section, consider revising instances of “should” to “could” when discussing recommendations for FDA

-Abstract (page 17): last sentence has a grammatical issue (“…discuss implications of this method finding for research and policy”). Perhaps the authors meant “methodological”?

-Literature Review (page 20): in the sentence “Substantially equivalence, however, proves only…”); I’d advise the authors to find an alternative word for “proves”. SE determinations are less of a tried-and-true fact, and more of a reflection of a regulatory assessment

-Literature Review (page 20): For the sentence “It is not uncommon for a device to cite a recalled predicate”, consider citing the new study by Everhart and colleagues in JAMA 2023 which estimates the prevalence of this phenomenon to be ~5% of all 510(k) devices

-Literature Review (page 21): Seems like there’s a typo in this sentence (“…systems, and found that (a cycle.7%) 510(k) clearances did not..”). Please fix.

-Literature Review (page 23): The authors use “database” as one word and “data bases” as two words in on this page. Please stick to just one word.

-Literature Review (page 22): When discussing FOIA requests, could reference FDA’s website for this process, which provides timelines for how long this takes. Could also reference studies from the literature showing the limitations of what information FDA is willing to redact

-Implications for Policy (page 38): “…”step” devices, which are submit and approved…”. “Submit” should be “Submitted”.

7. PLOS authors have the option to publish the peer review history of their article (what does this mean?). If published, this will include your full peer review and any attached files.

Reviewer #2: No

---

## [Author Response · Author response to Decision Letter 1]

20 Feb 2023

Thank you for giving us the opportunity to revise this paper. We are particularly grateful for the attention to detail of the reviewer, and willingness to look through a 103 page document! Below you will find a detailed response to yours and the reviewers’ comments. We have also included this as a separate document, in which our responses are in italics. This may be easier to follow. 

Substantive:

-The updated focus on product codes is nicely done, and a good addition to the literature. The revised Figure 2 captures this quite well in particular. Kudos.

Thank you again for your suggestion to change the focus of the paper. We agree that it improved the paper significantly.

-In the discussion, I think the authors should be clear that the limitations of their work are not simply to a specific type of device (Robotic Surgery devices), but also to a device category which is highly consolidated (limited competition – which means there are fewer available predicates that likely can be cited) and is by definition composed of multiple different devices (a similar issue for devices like orthopedic implants, but perhaps less of an issue for devices such as respiratory machines or endovascular catheters)

You are correct that this is a limit. We have added the following statement.

Moreover, given that for many years there was little or no competition to the Da Vinci Surgical Systems, we chose a device category for which there are likely to be fewer products available to cite as predicates than for products with higher levels of market competition.

-Introduction (page 18): the paragraph starting with “FDA has responded to these concerns…” could be deleted from the introduction. Some of this content is already in the discussion, and that is a more appropriate place for it. This change would also help shorten the introduction.

We agree and have taken out this paragraph

-Literature Review (page 23): Please revise this sentence: “The uniqueness of the device as an emerging technology makes it representative of the many challenges…”. It does not follow that a “unique” example can also be “representative”.

You are absolutely right, and the sentence was also a bit repetitive. We removed this sentence.

-When tracking product code divergence, did the authors also confirm that the change in product code was a decision by the manufacture, and not because of a reclassification order by FDA? As the latter would mean that it wasn’t so much an instance of “predicate creep” as it was updated regulation

We did not determine if the product classification code was made by the manufacturer or the FDA, or if it was driven by an administrative need or a technical feature. We now mention this in the limitations section. “Lastly, we did not determine if the product classification code was made by the manufacturer or the FDA, or if it was driven by an administrative reason or a technical feature.”

-Data Analysis (page 29): “This illustrates how larger jumps in the technological complexity of devices new devices can occur through the 510(k) process” – great summary statement. Note that there seems to be a typo, with “devices” repeated twice. Otherwise, this well captures the contributions of this study.

Thank you for catching this. It has been corrected.

-Discussion (page 32): The sentence “…to reduce barriers to bringing new medical devices” is not quite correct (that is perhaps a summation of the “Least Burdensome Principle”). The purpose of 510(k) is to facilitate incremental innovation in medical device development; not just reduce barriers. Please correct.

We have rewritten the sentence as “Given the purpose of the 510(k) process is to encourage incremental innovation in the medical device market by reducing regulatory barriers…..”

-Discussion (page 32): The sentence “..pre…”. “Guarantee” may not be appropriate here. The IOM report on 510(k) is clear that 510(k) clearance does not demonstrate safety. Would consider revising.

Thus, predicate creep in cases where the new device can be reasonably assured safe based on available scientific evidence, is beneficial to companies, patients, and regulators. 

-Discussion (page 33): “We were also able to see the absorption of a primary predicate device function into a secondary system function” – a really important interesting insight. Nicely done.\\

Thank you

-Limitations (page 35): In the sentence “scope of this research to robotic surgical devices”; it also may be useful to point about the fact about consolidation/monopolization of this space, which makes it harder to generalize to other product codes where there may be more device makers (and hence, more predicates)

We have added this sentence:

Moreover, given that for many years there was little or no competition to the Da Vinci Surgical Systems, we chose a device category for which there are likely to be fewer products available to cite as predicates than for products with higher levels of market competition.

-Implications for Research (Page 36): the point about patents is really interesting. Of note, there’s no good resource for this for device (e.g., there’s no equivalent of the FDA Orange Book for generic medicines as there is for devices).

That is true. We have noticed this when comparing research approaches with our research group’s work on pharmaceuticals 

-Implications for Policy (Page 38): The discussion about Step and Leap devices is interesting; I think the authors could provide more weight and clarity to this section by inverting the order of the current organization. First, highlight what FDA has already done, and then offer recommendations; bounces around a little too much right now.

Thank you for this suggestion. We have changed the order and it does make more sense now.

Stylistic

-The figures appear a bit blurry in my version of the PDF. Assuming Editors will work with the authors to ensure the figures in the final version of the paper are of a higher resolution

We agree. We will have to talk to the journal about this. The figures in our document are clear - but they change when uploaded into the system.

-At several points in the paper, the authors use the word “approved” (for example, page 21 of the Literature “These predicates can have also been approved…”). 510(k) does not provide “approvals”; the appropriate term is either “clearance” or “authorization”. Please revise all references throughout

Very good point. You are correct. We have changed all “approved” to “cleared”

-Throughout the article, the authors alternate from “Da Vinci” and “Da Vinci Si”. Please use just one consistent name.

Thank you for this suggestion. Except for cases where we are talking about the Da Vinci systems as a whole or another model, we have changed all mentions to Da Vinci Si

-In the Implications for Future Research section, consider revising instances of “should” to “could” when discussing recommendations for FDA

We have made this change

-Abstract (page 17): last sentence has a grammatical issue (“…discuss implications of this method finding for research and policy”). Perhaps the authors meant “methodological”?

We have changed this to read “We find that there is evidence of predicate creep using our method, and discuss implications of this method for research and policy.”

-Literature Review (page 20): in the sentence “Substantially equivalence, however, proves only…”); I’d advise the authors to find an alternative word for “proves”. SE determinations are less of a tried-and-true fact, and more of a reflection of a regulatory assessment

You are absolutely correct. We have changed this to read “Substantial equivalence, however, only supports an assessment that the device introduces no new safety hazards and functions at least as effectively as the predicate device.”

-Literature Review (page 20): For the sentence “It is not uncommon for a device to cite a recalled predicate”, consider citing the new study by Everhart and colleagues in JAMA 2023 which estimates the prevalence of this phenomenon to be ~5% of all 510(k) devices

Thank you for pointing out this article. I added this and another article in that same issue as references. 

-Literature Review (page 21): Seems like there’s a typo in this sentence (“…systems, and found that (a cycle.7%) 510(k) clearances did not..”). Please fix.

This has been fixed

-Literature Review (page 23): The authors use “database” as one word and “data bases” as two words in on this page. Please stick to just one word.

We have made this correction.

-Literature Review (page 22): When discussing FOIA requests, could reference FDA’s website for this process, which provides timelines for how long this takes. Could also reference studies from the literature showing the limitations of what information FDA is willing to redact

We have added a reference to the FDA time estimate to our discussion of FOIA, which is now in the conclusion. Since the issue of redacted data was not discussed in the paper, we did not add references regarding this issue. It now reads “We did consider requesting additional data through FOIA, but decided against this route given that it would add a significant amount of time to the data gathering, with an FDA estimate of 18 to 24 months [24], and there was a precedent in prior research to rely primarily on information provided in the FDA website [11]. 

-Implications for Policy (page 38): “…”step” devices, which are submit and approved…”. “Submit” should be “Submitted”.

This has been corrected

Thank you again for your detailed comments. It has significantly improved the quality of this paper.

---

## [Decision Letter · Decision Letter 2]

9 Mar 2023

Identification of predicate creep under the 510(k) process: A case study of a robotic surgical device

PONE-D-22-04558R2

Dear Dr. Sandra,

We’re pleased to inform you that your manuscript has been judged scientifically suitable for publication and will be formally accepted for publication once it meets all outstanding technical requirements.

Kind regards,

Quanzeng Wang

Academic Editor

PLOS ONE

Reviewers' comments:

Reviewer's Responses to Questions

**Comments to the Author**

1. If the authors have adequately addressed your comments raised in a previous round of review and you feel that this manuscript is now acceptable for publication, you may indicate that here to bypass the “Comments to the Author” section, enter your conflict of interest statement in the “Confidential to Editor” section, and submit your "Accept" recommendation.

Reviewer #2: All comments have been addressed

2. Is the manuscript technically sound, and do the data support the conclusions?

Reviewer #2: Yes

3. Has the statistical analysis been performed appropriately and rigorously? 

Reviewer #2: N/A

4. Have the authors made all data underlying the findings in their manuscript fully available?

Reviewer #2: Yes

5. Is the manuscript presented in an intelligible fashion and written in standard English?

Reviewer #2: Yes

6. Review Comments to the Author

Reviewer #2: Thank you very much for the detailed and responsive edits; I appreciate the engagement with the reviewer & editorial feedback; and believe the manuscript will be an important contribution to the literature. No further comments from me. Congratulations to the authors.

7. PLOS authors have the option to publish the peer review history of their article (what does this mean?). If published, this will include your full peer review and any attached files.

Reviewer #2: No

---

## [Editor Report · Acceptance letter]

17 Mar 2023

PONE-D-22-04558R2 

Identification of predicate creep under the 510(k) process: A case study of a robotic surgical device 

Dear Dr. Rothenberg:

I'm pleased to inform you that your manuscript has been deemed suitable for publication in PLOS ONE. Congratulations! Your manuscript is now with our production department. 

Kind regards, 

on behalf of

Dr. Quanzeng Wang 

Academic Editor

PLOS ONE